# Image Guided Visual Tracking Control System for Unmanned Multirotor Aerial Vehicle with Uncertainty †

**Shafiqul Islam [1],*, Husameldin Mukhtar [2] and Jorge Dias [3]**

[1]  Department of Computer Science and Engineering, NCF-235 Academic Science Complex, 1 Drexel Drive, Xavier University of Louisiana, New Orleans, LA 70125, USA

[2]  Department of Electrical Engineering, University of Dubai, Dubai 14143, UAE; hhadam@ud.ac.ae

[3]  Institute of Systems and Robotics, University of Coimbra, 3030-260 Coimbra, Portugal; jorge@deec.uc.pt

*   Correspondence: sislam3@xula.edu; Tel.: +1-504-520-5217

†   This paper is an extended version of our paper published in Islam, S.; Mukhtar, H.; Al Khawli, T.; Sunda-Meya, A. Wavelet-Based Visual Tracking System for Miniature Aerial Vehicle. In Proceedings of the 44th Annual Conference of the IEEE Industrial Electronics Society (IECON), Omni Shoreham Hotel, Washington, DC, USA, 21–23 October 2018; pp. 2651–2656.

**Abstract:** This paper presents a wavelet-based image guided tracking control system for unmanned multirotor aerial vehicle system with the presence of uncertainty. The visual signals for the visual tracking process are developed by using wavelet coefficients. The design uses a multiresolution interaction matrix with half and details images to relate the time-variation of wavelet coefficients with the velocity of the aerial vehicle and controller. The proposed design is evaluated on a virtual quadrotor aerial vehicle system to demonstrate the effectiveness of the wavelet-based visual tracking system without using an image processing unit in the presence of uncertainty. In contrast to the classical visual tracking technique, the wavelet-based method does not require an image processing task.

**Keywords:** unmanned aerial vehicle (UAV); visual tracking system; control; multiresolution wavelet transform (MWT)

## 1. Introduction

Over the past years, there have been tremendous research interest on the development of autonomous flight tracking system for small scale multirotor miniature aerial vehicle. This demands may be because of its wide variety of civilian and military applications. The literature review on control designs on this area can be found in [1]. The survey on small scale aerial vehicle can be traced from [2–4]. Some textbooks have also been reported on this area, for example, [5–8]. The most recent results in this area can also be found here [9,10]. The image guided tracking control design for the vehicle has also been studied by researchers and industrial, for example [11–30]. In classical vision-based tracking control design, one usually uses visual/geometrical information for example image points, lines and circles that obtained from either one or multiple cameras to minimize the error between a set of actual and reference measurements. These vision based tracking designs require to process images which associated with features extraction and matching tasks over time. The goal in image processing is to identify and match the geometrical features in the image segments [28,31]. This segmented process is computationally expensive slowing down the tracking process significantly. The integrating of additional algorithms slows down the tracking convergence speed significantly. To deal with these problems, a new type of image-based tracking methods have recently been proposed. The results in this area can be traced from [31–36]. Authors in [31] showed that vision-based control

can be designed without using the image processing process. In [32,33], authors presented visual tracking mechanism by using pure photometry image signal. Authors in [34] developed visual tracking approaches based on using image gradient. The entropy-based visual tracking technique also introduced in [35]. Recently, authors in [36] designed visual tracking schemes for automatic positioning under a scanning electron microscope by using Fourier transformation. These methods may be used to relax the requirement of the image processing system while ensuring accurate visual tracking without the presence of uncertainty. Most recently, authors in [37] applied a wavelet-based visual tracking system for rigid robot manipulators. Recently, authors in [36] designed visual tracking schemes for automatic positioning under a scanning electron microscope by using Fourier transformation. These methods can be used to relax the requirement of the image processing system while ensuring accurate visual tracking without the presence of uncertainty. Most recently, authors in [37] applied a wavelet-based visual tracking system for rigid robot manipulators. Most recently, authors in [38] presented a wavelet-based tracking system for an aerial vehicle. However, the design and analysis assumed that the vision and vehicle system is free from uncertainty. Moreover, the design and analysis do not provide the convergence analysis of the closed-loop system formulated by visual and vehicle tracking control algorithm. In our view, the visual tracking process and control algorithms in many of these above methods may deteriorate significantly with the presence of uncertainty associated with the visual/image processing and modeling errors, control inputs, and other external disturbances including flying environments.

This paper presents a wavelet-based image guided tracking control system for the multirotor aerial vehicle in the presence of uncertainty. The proposed wavelet based design can also be used in other areas such as corners detection, pattern recognition, filtering, economic data and data compression, compressed sensing, and temperature analysis. In contrast with the Fourier transform, the wavelet transform can be applied to express the image in both the time and frequency domain. The wavelet design uses a multiresolution wavelet transform (MWT) to develop an image-guided tracking control algorithm for the multirotor aerial vehicle. In our design, we use a half-resolution image obtained from the MWT considered as a visual signal. The design develops the spatial derivative wavelet coefficients involved in computing multiple resolution interaction matrices relating to the time-variation of derivative wavelet coefficients with vehicles' spatial velocity from the detail wavelet coefficients. The half-resolution based design can provide automatic filtering of the low and high frequencies in the image generally corresponding to the image noise. Such filtering allows selecting noiseless and redundant visual signals for a more accurate and stable visual tracking control system. The proposed design is extensively evaluated on a virtual quadrotor aerial vehicle platform with half and details image-based MWT in the presence of uncertainty. The tests are conducted in nominal conditions and using different coefficients resolution to express the optimal ones improving the controller behaviors concerning convergence, robustness, and accuracy. The evaluation results show that the MWT based design can provide accuracy and efficiency without using image processing tasks with the presence of uncertainty.

The paper is organized as follows: Section 2 reviews the basics of the MWT and the dynamical model of the systems. Section 2 also presents the design and analysis of the MWT based tracking control system for a multirotor aerial vehicle. Section 3 presents design synthesis and test results on a quadrotor aerial vehicle system. Finally, Section 4 provides concluding remarks of this work.

## 2. Wavelet-Based Visual Tracking System Design for UAV

The visual tracking system has been extensively studied by many researchers from computer vision and image processing communities. The design allows the system to perceive the surrounding environment, make decisions, and react to changes relying on interdisciplinary research paradigm including computer vision, image processing, and control system. Classical visual tracking usually involves image processing task to detect and match the geometrical features in the image. The image processing task affects the tracking performance significantly as the process requires high

computational effort and slows down the tracking control system. To deal with this problem, this work focuses on the design and development of a wavelet-based visual tracking system. First, we present a brief background review on the basics of the multiresolution wavelet transformation mechanism. Then, we introduce a wavelet-based visual tracking system by developing a model and visual control strategy. Finally, a dynamical model and visual tracking system for a quadrotor unmanned aerial vehicle system are presented.

### 2.1. Multiresolution Wavelet Transform (MWT)

Wavelets transform is a mathematical tool that provides a representation of signals in time and frequency domains [39]. Such transformation aims to decompose the original full resolution image into an approximation half resolution image, a horizontal, vertical, and diagonal details images [38,40–42].

We consider a 2D signal $\mathcal{F}(x,y)$ and a wavelet function $\mathcal{G}(x,y)$. Their inner products are defined as the general wavelet transform as

$$\left\langle \mathcal{F}(x,y), \mathcal{G}(x,y) \right\rangle = \int_{-\infty}^{+\infty} \int_{-\infty}^{+\infty} \mathcal{F}(x,y), \mathcal{G}(x,y) dx dy \tag{1}$$

For MWT, two functions have to be defined first, a scaling function and a mother wavelet function. The scaling function ($\Phi$) can be modeled as a low pass filter with certain Daubechies pair of the fourth-order coefficients. The mother wavelet ($\Psi$) can be modeled as a high pass filter with different coefficients [43]. MWT is designed by defining four different combinations related to the wavelet function $\mathcal{G}(x,y)$ to generate four different subsignals. The four combined signals can be designed as

$$\begin{align}
\Gamma^O(x,y) &= (\Delta \circ \Phi(x)) \circ (\Delta \circ \Phi(y)) \tag{2} \\
\Gamma^H(x,y) &= (\Delta \circ \Phi(x)) \circ (\Delta \circ \Psi(y)) \tag{3} \\
\Gamma^V(x,y) &= (\Delta \circ \Psi(x)) \circ (\Delta \circ \Phi(y)) \tag{4} \\
\Gamma^D(x,y) &= (\Delta \circ \Psi(x)) \circ (\Delta \circ \Psi(y)) \tag{5}
\end{align}$$

where $\Delta$ is the down-sampling operator and $\circ$ refers to the composition of functions. In this paper, the operator $\Delta$ will be referred as $\downarrow 2$. Finally, the original full-resolution image $I_{(2j+1)}$ is decomposed into four subsignals through the inner product with each defined by four combinations as

$$I_{(2j)} = \left\langle I_{2^{(j+1)}}, \Gamma_{2j}^O(x,y) \right\rangle \tag{6}$$

to achieve the approximation image $I_{(2j)}$ at half resolution, and

$$d_{(2j)}^k = \left\langle I_{\left(2^{(j+1)}\right)}, \Gamma_{2j}^k(x,y) \right\rangle \tag{7}$$

$\forall k = H, V, D$ to achieve the horizontal, vertical, and diagonal details.

For a better understanding, Equations (6) and (7) can be summarized as depicted in Figure 1. Specifically, an example of applying low-pass and high-pass filters in the rows and columns directions according to the image given in Figure 2 is illustrated in Figure 3.

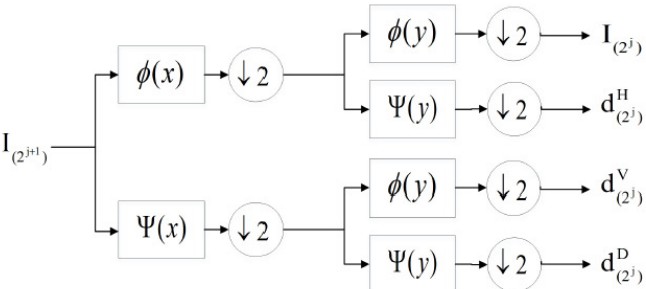

**Figure 1.** Multiresolution wavelet transform.

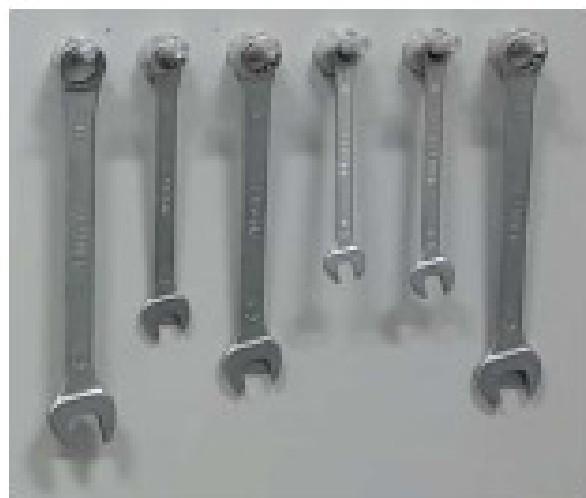

**Figure 2.** Original image.

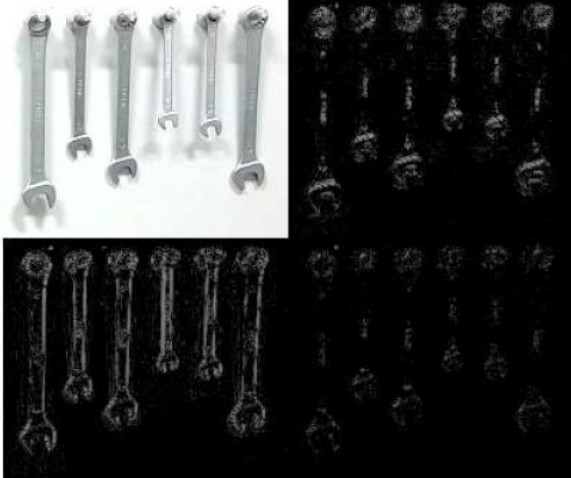

**Figure 3.** MWT approximation image, horizontal details, vertical details, and diagonal details.

*2.2. MWT Based Interaction Matrix Modeling*

Let us first model the change in camera position to the change in features before developing a wavelet-based autonomous visual tracking system. The relationship between the camera movement and the corresponding variation of the detected features can be described by the following model

$$\dot{s} = L_s v_c \tag{8}$$

where $L_s$ is the interaction matrix (or the feature Jacobian) that links the change in camera position to the change in features, and $v_c$ is the camera velocity vector including instantaneous linear and angular velocities. In wavelet-based based design, the half resolution approximation image is used as the visual signal $s = I_{(2^j)}$. Specifically, the luminance $I_x$ at location $x = (x, y)$ is considered to be the new feature. For that purpose, a new interaction matrix related to the luminance should be estimated as given by the following equation

$$\dot{I}_x(t) = L_{I_x} v_c \tag{9}$$

Now, consider a 3D point $P(t)$ in the world being projected to the image plane as $p(t)$. The variation in $P(t)$ may occur either because of the object motion or camera motion. The relative motion estimation between two images can be illustrated using optical flow [44]. Since (9) requires finding the change in luminance, one can estimate the luminance in the following form

$$\dot{I}(p(t), t) = \Delta I\,(p(t), t)^T\,\dot{p}(t) + \frac{\partial I(p(t), t)}{\partial t} \tag{10}$$

However, if at time $t$, the normalized coordinates of the point $p(t)$ coincide with $x$, then Equation (10) can be written as

$$\dot{I}(p(t), t) = \nabla I_x(t)^T \dot{x} + \dot{I}_x(t) \tag{11}$$

where $\Delta I_x(t)$ is the spatial gradient of $I_x(t)$, and $\dot{x}$ is the 2D velocity of the point $p(t)$. Following the luminance constancy assumption, then substitute $\dot{I}(p(t), t) = 0$, and Equation (10) becomes

$$\nabla I\dot{x} + \frac{\partial I}{\partial t} = 0 \tag{12}$$

Using Equation (6) and (7) and mathematical steps presented in [45], one has

$$\left\langle I_{(2^{j+1})}, \frac{\partial \Gamma^H}{\partial x} \right\rangle \dot{x} + \left\langle I_{(2^{j+1})}, \frac{\partial \Gamma^V}{\partial y} \right\rangle \dot{y} + \frac{\partial}{\partial t} \left\langle I_{(2^{j+1})}, \Gamma^O \right\rangle = 0 \tag{13}$$

For simplicity, the following two functions are introduced

$$g_{(2j)}^H \cong \left\langle I_{(2^{j+1})}, \frac{\partial \Gamma^H}{\partial x} \right\rangle, g_{(2j)}^V \cong \left\langle I_{(2^{j+1})}, \frac{\partial \Gamma^V}{\partial y} \right\rangle \tag{14}$$

where $g_{(2j)}^H$ and $g_{(2j)}^V$ are the derivative horizontal and vertical details as illustrated in Figure 4. Now, by substituting (6) and (14) in (13), one can write

$$g_{(x,y)}^H \dot{x} + g_{(x,y)}^V \dot{y} + \frac{\partial}{\partial t} I_{(x,y)} = 0 \tag{15}$$

Using feature as $s = I_{(2j)}$, we have

$$\left[ g_{(x,y)}^H \;\; g_{(x,y)}^V \right] \dot{x} + \dot{s}(x, y) = 0 \tag{16}$$

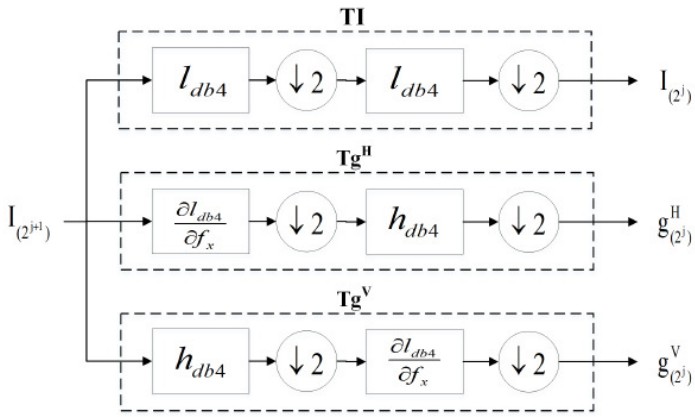

**Figure 4.** MWT showing the derivative horizontal and vertical details.

Now, using the relation between the change in features and the camera velocity for the 2D feature and $\dot{s} = L_s v_c$ with

$$L_s = \begin{bmatrix} \frac{-1}{Z} & 0 & \frac{x}{Z} & xy & -(1+x^2) & y \\ 0 & \frac{-1}{Z} & \frac{y}{Z} & 1+y^2 & -xy & -x \end{bmatrix} \tag{17}$$

one can write $\dot{s}$ as

$$\dot{s} = - \begin{bmatrix} g^H_{(2^j)} & g^V_{(2^j)} \end{bmatrix} L_s v_c \tag{18}$$

Equation (18) can be simplified as

$$\dot{s} = L_w v_c \tag{19}$$

where $L_w$ is the new multiresolution interaction matrix $L_w = - \begin{bmatrix} g^H_{(2^j)} & g^V_{(2^j)} \end{bmatrix} L_s$.

*2.3. Visual Tracking Control System Design and Stability Analysis*

Our aim is now to develop wavelet based visual tracking system. The goal is to generate velocities aiming to decrease the error exponentially. For the case of wavelets-based based design, the tracking error is the difference between the current and the desired approximation images in each iteration as expressed as

$$e = I_{2j} - I_{2j}^* \tag{20}$$

Then, the relationship between the camera movement and the corresponding variation of detected features can be described by the following model

$$\dot{s} = \dot{x} = L_w v_c \tag{21}$$

Now, a similar model that maps the camera velocity with the variation of the error through the newly designed multiresolution interaction matrix can be described as

$$\dot{e} = L_w v_c \tag{22}$$

To ensure an exponential convergence of the tracking error $\dot{e} = -\lambda e$, one can design image guided wavelet based velocity tracking controller as

$$v_c = -\lambda L_w^+ e \tag{23}$$

where $L_w{}^+$ is the Moore-Penrose pseudoinverse of the multiresolution interaction matrix. If the interaction matrix $L_w$ has a full rank, then we have $L_w{}^+ = \left(L_w{}^T L_w\right)^{-1} L_w{}^T$. This implies that the signals $v_c$ and $\left(\dot{e} - \lambda L_w l_w{}^+ e\right)$ are bounded and guaranteed to be minimum values. When interaction matrix $L_w$ has a full rank and $det(L_w) \neq 0$, then the matrix $L_w$ is invertible. This means that it is possible to ensure the existence of the velocity control $v_c = -\lambda L_w^{-1} e$. However, in practice, the interaction matrix $L_w$ or $L_w^+$ is usually approximated to reduce the computational complexity. In practical application, the interaction matrix or its pseudoinverse are usually estimated or approximated as it may not be possible to find their precise values. As a result, the image guided wavelet based autonomous visual tracking control needs to change to the following form

$$v_c = -\lambda \widehat{L_w}^+ e \tag{24}$$

where $\widehat{L_w}^+$ denotes the approximated pseudoinverse multiresolution interaction matrix. The interaction matrix is approximated to reduce the computational complexity. For a particular task, the interaction matrix is computed at the desired location and the result is used for all iterations $\hat{L}_w = L_w^*$. An additional step is used to achieve a smooth convergence of the tracking task defined as the Levenberg–Marquardt optimization technique [9]. So, the new optimized visual tracking control law can be written in the following form

$$v_c = -\lambda \left(\mu I_{6\times 6} + \widehat{L_w}^T \widehat{L_w}\right)^{-1} \widehat{L_w}^+ \left(I_{2j} - I_{2j}^*\right) \tag{25}$$

The final block diagram of the multiresolution wavelet based visual tracking control system is illustrated in Figure 5 with all the details. It shows how the half-resolution approximation image is used as the visual signal and how the derivative horizontal and vertical details are used to modify the new multiresolution interaction matrix before being used in the new optimized tracking control law. The signals $TI$, $Tg^H$ and $Tg^V$ in Figure 5 are presented in Figure 4. Based on our above analysis, one can state that the linear and angular velocity tracking error signals of the camera are bounded and exponential converge to zero. Since the tracking error velocity signals are bounded, the linear and angular position tracking error signals is also bounded and exponentially converge to zero.

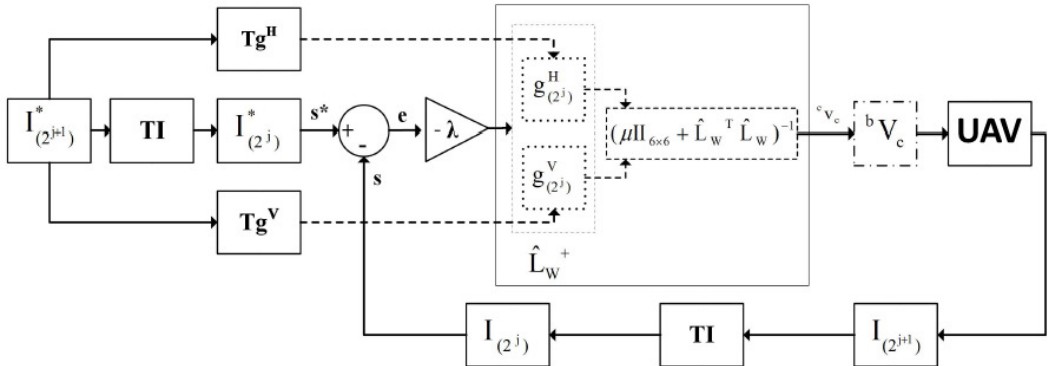

**Figure 5.** Detailed block diagram of wavelet based visual tracking system.

## 2.4. Dynamical Model for UAV

Let us now introduce the dynamical model of the quadrotor aerial vehicle by Using Newton–Euler formulation. The dynamical model for the vehicle can be established with the presence of uncertainty as [10,38]

$$\begin{aligned}
m\ddot{x} &= (\cos\phi\sin\theta\cos\psi + \sin\phi\sin\psi)\,u_1 + D_x \\
m\ddot{y} &= (\cos\phi\sin\theta\sin\psi - \sin\phi\cos\psi)\,u_1 + D_y \\
m\ddot{z} &= \cos\phi\cos\theta u_1 - mg + D_z
\end{aligned} \tag{26}$$

$$\begin{aligned}
I_x\ddot{\phi} &= \dot{\phi}\dot{\psi}\left(I_y - I_z\right) + J_r\dot{\theta}\Omega_r + lu_2 + D_\phi \\
I_y\ddot{\theta} &= \dot{\theta}\dot{\psi}\left(I_z - I_x\right) - J_r\dot{\phi}\Omega_r + lu_3 + D_\theta \\
I_z\ddot{\psi} &= \dot{\phi}\dot{\theta}\left(I_x - I_y\right) + J_r\Omega_r + u_4 + D_\phi
\end{aligned} \tag{27}$$

where $m \in \Re$ and $(I_x, I_y, I_z) \in \Re$ denotes the mass and moment of inertia, $(x, y, z)$ and $(\phi, \theta, \psi)$ presents the position and rotational angles defining the roll, pitch and yaw, respectively, $u_1$ defines the thruster input force vector, $u_2$, $u_3$ and $u_4$ describes the control inputs for the roll, pitch and yaw moments, $g$ is the gravitational force, $l$ denotes the length from the center of the mass of the vehicle and $D_x$, $D_y$, $D_z$, $D_\phi$, $D_\theta$ and $D_\psi$ defines the lumped uncertainty associated with mass, inertia, aerodynamic friction, and other external disturbances including flying environment. The dynamical model for the quadrotor aerial vehicle system can also be written in state-space form as [10]

$$\begin{aligned}
\dot{x}_1 &= x_2, \dot{x}_2 = au_{fx}u_1 + \mu_x, \dot{x}_3 = x_4, \dot{x}_4 = au_{fy}u_1 + \mu_y \\
\dot{x}_5 &= x_6, \dot{x}_6 = a\cos(y_1)\cos(y_3)u_1 - g + \mu_z
\end{aligned} \tag{28}$$

$$\begin{aligned}
\dot{y}_1 &= y_2, \dot{y}_2 = bu_2 + \mu_\phi + b\zeta_\phi, \dot{y}_3 = y_4, \dot{y}_4 = cu_3 + \mu_\theta + c\zeta_\theta \\
\dot{y}_5 &= y_6, \dot{y}_6 = du_4 + \mu_\theta + d\zeta_\psi
\end{aligned} \tag{29}$$

with $x_1 = x$, $x_3 = y$, $x_5 = z$, $y_1 = \phi$, $y_3 = \theta$, $y_5 = \psi$, $\mu_x = aD_x$, $\mu_y = aD_y$, $\mu_z = aD_z$, $\mu_\phi = bD_\phi$, $\mu_\theta = cD_\theta$, $\mu_\psi = dD_\psi$, $a = m^{-1}$, $b = \frac{l}{I_x}$, $c = \frac{l}{I_y}$, $d = \frac{1}{I_z}$, $\zeta_\phi = y_2y_6(I_y - I_z) + J_ry_4\Omega_r$, $\zeta_\theta = y_4y_6(I_z - I_x) + J_ry_2\Omega_r$, $\zeta_\psi = y_2y_4(I_x - I_y) + J_r\Omega_r$, $u_{fx} = (\cos\phi\sin\theta\cos\psi + \sin\phi_i\sin\psi)$ and $u_{fy} = (\cos\phi\sin\theta\sin\psi - \sin\phi\cos\psi)$. The proposed design and stability analysis is based on the following assumptions: $A_1$: The attitude angles are bounded as $-\frac{\pi}{2} < \phi < \frac{\pi}{2}$, $-\frac{\pi}{2} < \theta < \frac{\pi}{2}$ and $-\pi < \psi < \pi$. $A_2$: The terms $\mu_x$, $\mu_y$, $\mu_z$, $\mu_\phi$, $\mu_\theta$, $\mu_\psi$ are continuous and bounded by known constants belonging to a compact sets.

### 2.5. Tracking Control System Design and Stability Analysis

Let us now develop a wavelet based visual flight tracking control system for the attitude, altitude, and virtual position dynamics of the quadrotor aerial vehicle system. For the sake of simplicity, the position and velocity states of the vehicle are defined as $x_v = [x_1, x_3, x_5, y_1, y_3, y_5]$, $\dot{x}_v = [x_2, x_4, x_6, y_2, y_4, y_6]$. The states of the camera are defined as $x_T = [x_{1c}, x_{3c}, x_{5c}, y_{1c}, y_{3c}, y_{5c}]$ and $v_T = [x_{2c}, x_{4c}, x_{6c}, y_{2c}, y_{4c}, y_{6c}]$ with $x_{1c} = x^c$, $x_{3c} = y^c$, $x_{5c} = z^c$, $y_{1c} = \phi^c$, $y_{3c} = \theta^c$, $y_{5c} = \psi^c$, $x_{2c} = v^c_x$, $x_{4c} = v^c_y$, $x_{6c} = v^c_z$, $y_{2c} = v^c_\phi$, $y_{4c} = v^c_\theta$, $y_{6c} = v^c_\psi$, In our analysis, the goal is to show that the states of the vehicle converge to the desired camera states asymptotically provided that the position and velocity states of the camera are bounded and exponentially converge to zero.

In design and analysis, the model parameters of the vehicle are assumed to be constant. The bounds of the disturbances are known and lie over the given compact sets. Then, because of assumption $A_1$ and $A_2$, it is possible to develop a control algorithm such that the visual tracking system can ensure the bound of the position and velocity states of the vehicle $x_v$ and $\dot{x}_v$ provided that the desired position and velocity of the camera $x_T$ and $v_T$ are bounded as derived in the previous subsection. The design considers that the attitude dynamics are fully actuated and linearised by decoupling the first three terms in the model (27). The design also considers that the attitude dynamic

is faster than the translation dynamics. Now, we introduce the following visual tracking control algorithm for the thruster input $u_1$

$$u_1 = \frac{g}{a\cos(y_1)\cos(y_3)} - k_{dz}e_6 \tag{30}$$

where $k_{dz} > 0$, $e_6 = (x_{6c} - x_6)$ and $x_{6c}$ is the desired linear velocity of the camera-generated by the $z$ components of the camera velocity vector. The vector $u_1$ is the desired thruster in $z$ direction to the vehicle. Now, the goal is to develop wavelet-based image-guided attitude control laws $u_2$, $u_3$, and $u_4$ for the quadrotor vehicle. The roll and pitch control laws are used to control the translation to $x$, $y$, and $z$ axis, respectively. To design $u_2$, $u_3$, and $u_4$, the desired rolling, pitching, and yawing angles, as well as their angular rates, are required. The desired angular rates of the rolling and pitching angles are obtained from the fourth, fifth, and sixth components of the camera velocity $v_T$. The desired angular position of the rolling and pitching angles is developed by using the following relationship

$$y_{1d} = arcsin\left(u_{tx}\right), y_{3d} = arcsin\left(u_{ty}\right) \tag{31}$$

where the virtual input algorithms $u_{tx}$ and $u_{ty}$ are designed as

$$u_{tx} = -\hat{k}_{dx}e_2, u_{ty} = -\hat{k}_{dy}e_4 \tag{32}$$

with $\hat{k}_{dx} > 0$, $\hat{k}_{dy} > 0$, $e_2 = (x_{2c} - x_2)$ and $e_4 = (x_{4c} - x_4)$ Then, attitude control algorithms for the rolling, pitching and yawing moments can be designed as

$$u_2 = -k_{p\phi}e_7 - k_{d\phi}e_8, u_3 = -k_{p\theta}e_9 - k_{d\theta}e_{10}, u_4 = -k_{d\psi}e_{12} \tag{33}$$

with $k_{p\phi} > 0$, $k_{d\phi} > 0$, $k_{p\theta} > 0$, $k_{d\theta} > 0$, $k_{d\psi} > 0$, $e_7 = (y_{1d} - y_1)$, $e_9 = (y_{3d} - y_3)$ and $e_{12} = (y_{6c} - y_6)$. To show tracking error convergence with controller (30)–(33), we first derive the tracking error model in the following state space equation

$$\dot{e}_1 = e_2, \dot{e}_2 = \dot{x}_{2c} - \eta_x, \dot{e}_3 = e_4, \dot{e}_4 = \dot{x}_{4c} - \eta_y \tag{34}$$

$$\dot{e}_5 = e_6, \dot{e}_6 = \dot{x}_{6c} - \eta_z, \dot{e}_7 = e_8, \dot{e}_8 = \dot{y}_{2d} - \eta_\phi \tag{35}$$

$$\dot{e}_9 = e_{10}, \dot{e}_{10} = \dot{y}_{4d} - \eta_\theta, \dot{e}_{11} = e_{12}, \dot{e}_{12} = \dot{y}_{6c} - \eta_\psi \tag{36}$$

where $e_1 = (x_{1c} - x_1)$, $e_2 = (x_{2c} - x_2)$, $e_3 = (x_{3c} - x_3)$, $e_4 = (x_{4c} - x_4)$, $e_5 = (x_{5c} - x_5)$, $e_6 = (x_{6c} - x_6)$, $e_7 = (y_{1d} - y_1)$, $e_8 = (\dot{y}_{1d} - y_2)$, $e_9 = (y_{3d} - y_3)$, $e_{10} = (\dot{y}_{3d} - y_4)$, $e_{11} = (y_{5c} - y_5)$, $e_{12} = (y_{6c} - y_6)$, $\eta_x = \left(au_{fx}u_1 + d_x\right)$, $\eta_y = \left(au_{fy}u_1 + d_y\right)$, $\eta_z = a\cos(y_1)\cos(y_3)u_1 - g + d_z$, $\eta_\phi = bu_2 + d_\phi$, $\eta_\theta = cu_3 + d_\theta$, $\eta_\psi = du_4 + d_\theta$ with $b\zeta_\phi = 0$, $c\zeta_\theta = 0$, $d\zeta_\psi = 0$. The Lyapunov method is used to show the convergence of the closed loop system. To do that, the following Lyapunov function candidate is chosen as $V_T = V_x + V_y + V_z + V_\phi + V_\theta + V_\varphi$ where $V_o = \frac{e_m^T e_m}{2}$ with $o = x, y, z, \phi, \theta, \varphi$ and $m = 2, 4, 6, 8, 10, 12$. In view of assumptions $A_1$ and $A_2$ and using the inequality $cA^T B \leq \delta_c A^T A + c^{-1}B^T B$ with $\delta_c > 0$, the time derivative of $V_T$ along the closed loop tracking error models formulated by (30)–(36) can be simplified as $\dot{V}_T \leq 0$. This implies that there exist control design parameters such that the visual tracking control algorithms (30) to (33) can ensure that the states of the vehicle $x_v$ and $\dot{x}_v$ are bounded and asymptotically stable provided that the position and velocity of the camera $x_T$ and $v_T$ are bounded and exponentially converge to zero as derived in the previous subsection. This implies that all the error states signals in closed-loop tracking error systems are bounded and asymptotically stable in the Lyapunov sense.

## 3. Design Synthesis and Test Results

This section presents the test results of the proposed wavelet-based visual tracking system on a quadrotor aerial vehicle. The implementation block diagram of the wavelet-based autonomous visual tracking system is depicted in Figure 6.

The test is conducted on a virtual quadrotor aerial vehicle system. The vehicle is equipped with a downward camera attached to the center of the vehicle. The camera is equipped with the vehicle such that it can look down to the vehicle. In our test, we use a known image as the reference target object which estimates the position (3D) of the vehicle. The target is synchronized in such a way that its axes are parallel to the local plane North East axes. First, we calibrate the camera before conducting a test to calculate the camera intrinsic parameters. The focal length and sensor size are selected as $f = 3$ (mm) and $p = 10(\mu m)$. Figure 7 shows the decomposed reference image into horizontal, vertical, approximation, and diagonal coefficients. For our test, the width and height of the image are selected as $W = 256$ and $H = 256$ Pixels. The initial linear position of the vehicle, camera and rotation is defined as $T_x = 0.02$ (m), $T_y = 0.02$ (m), $T_z = 0.7$ (m), $R_x = 5(^o)$, $R_y = 5(^o)$, and $R_z = 10(^o)$. The initial camera image is shown in Figure 8. Figure 9 depicts the initial image with the reference image. Using the proposed MWT based wavelet based visual tracking design, the final image is shown in Figure 10. Figure 11 presents the error image between the final image and the reference image. The error norm of the MWT based visual tracking system is depicted in Figure 12. Let us now select the parameters for the evaluated quadrotor aerial vehicle as $I_x = 3.8278 \times 10^{-3}$, $I_y = 3.8288 \times 10^{-3}$, $I_z = 7.6566 \times 10^{-3}$, $m = 0.48$ (kg), $l = 0.25$ (m), $g = 9.8$ (m/s), $J_r = 2.8385 \times 10^{-5}$ and $\Omega_r = 3.60 \times 10^3$.

The controller design parameters for the aerial vehicle are selected as $\hat{k}_{dx} = 0.1$, $\hat{k}_{dy} = 0.1$, $k_{dz} = 5$, $k_{d\phi} = 0.5$, $k_{d\theta} = 1$, $k_{d\psi} = 1$, $kp_\phi = 5$ and $kp_\theta = 5$. The controller parameters for the visual tracking controller are selected as $\lambda = 0.3$ and $\mu = 3 \times 10^{-3}$. The sampling time for all tests is chosen as 0.01 (s). In our test, the camera position with respect to vehicle $T_c^v$ is assumed to be known and constant. Figure 13 depicts the coordinate system of the vehicle, camera, and target object. Figure 14 presents the image of the real scene of the flying vehicle interacting with target objects. To test the wavelet-based based design on a quadrotor vehicle, we first generate a vehicle's position matrix to the ground $T_v^g$ by using a homogeneous transformation matrix $T_h$.

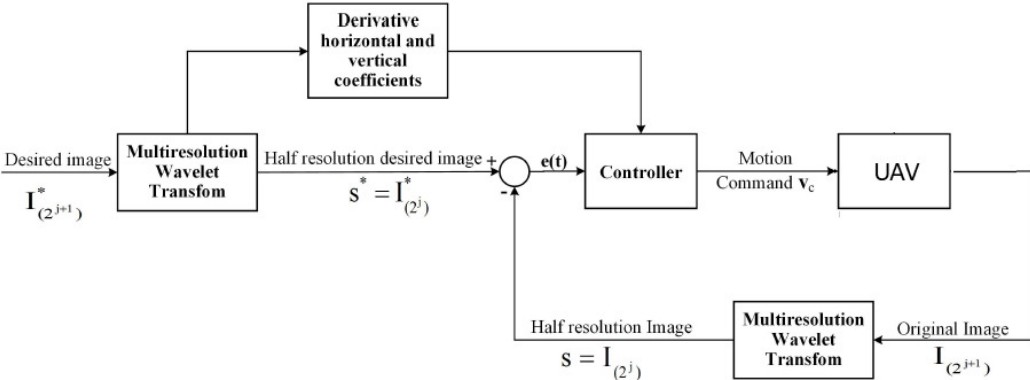

**Figure 6.** Implementation block diagram of the wavelet based visual tracking system.

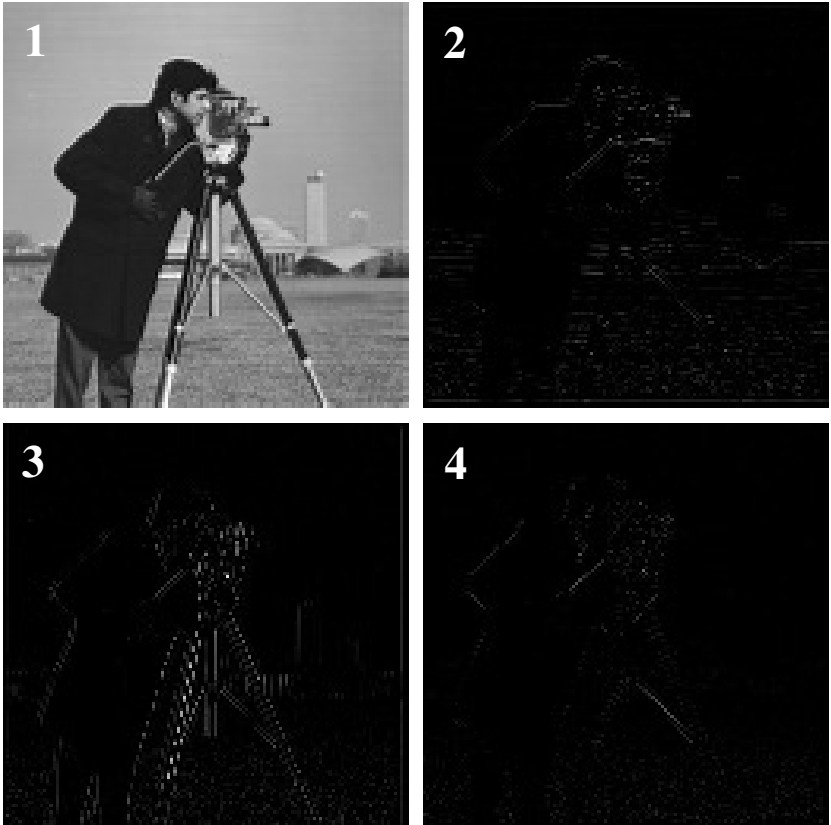

**Figure 7.** The wavelet decomposition of the reference image approximation-horizontal, vertical, and diagonal coefficients [38] (© 2018 IEEE).

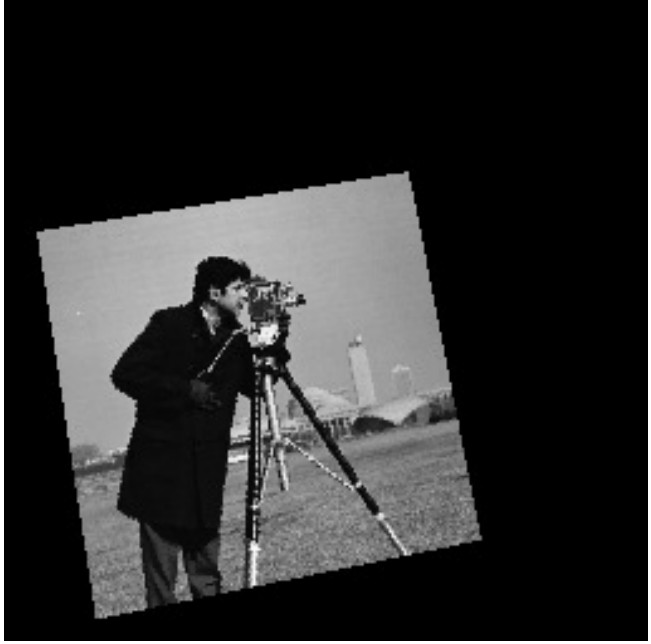

**Figure 8.** The initial image uses in our test [38] (© 2018 IEEE).

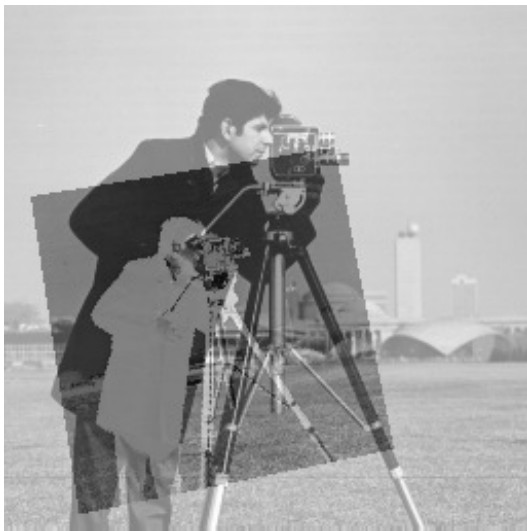

**Figure 9.** The initial and reference image [38] (© 2018 IEEE).

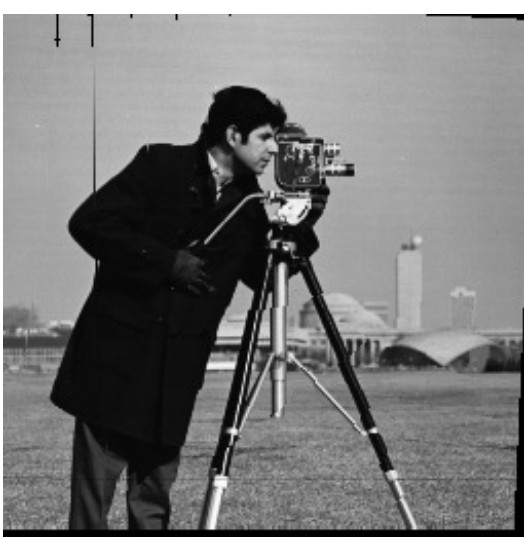

**Figure 10.** The final image based on using MWT based visual tracking system [38] (© 2018 IEEE).

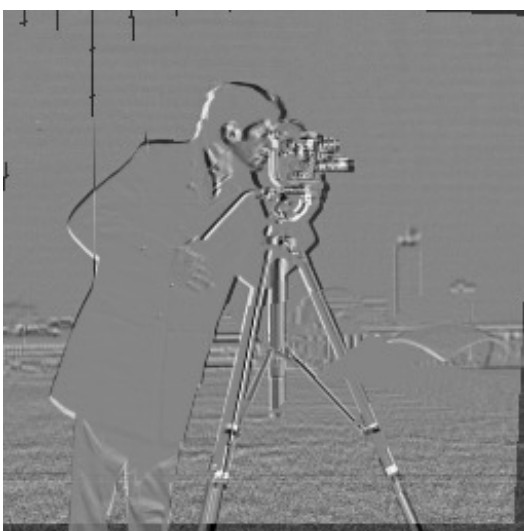

**Figure 11.** The final image difference with respect to the reference image [38] (© 2018 IEEE).

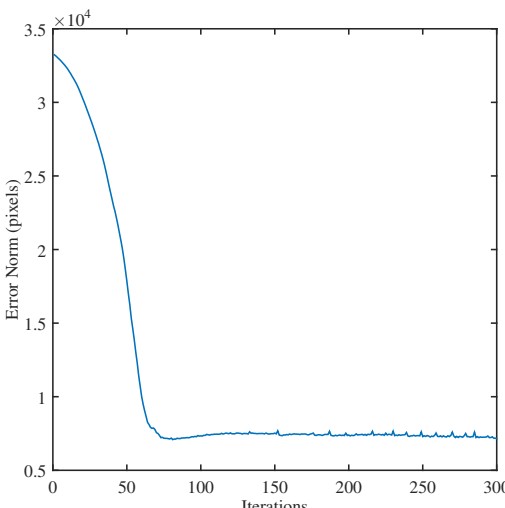

**Figure 12.** The profile of the error norm of the wavelet based visual tracking system [38] (© 2018 IEEE).

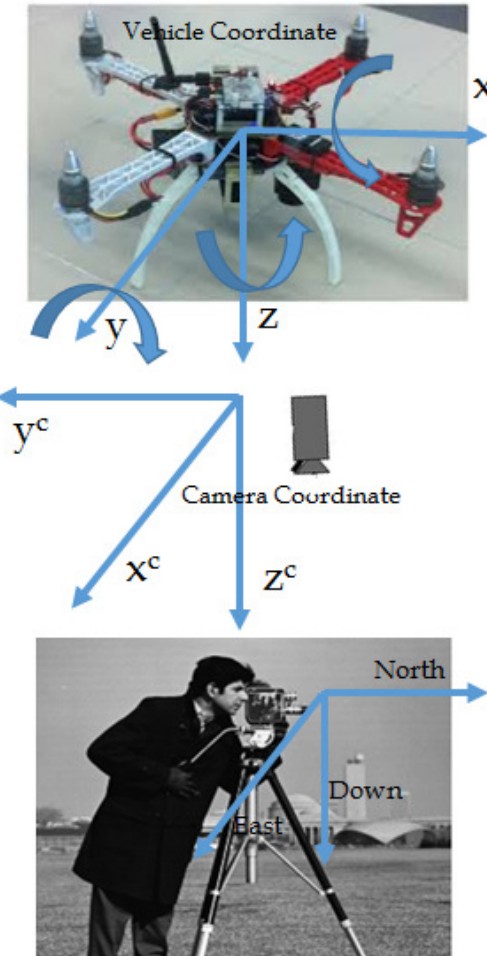

**Figure 13.** The coordinate systems-vehicle, camera and target object.

The vehicles position states are used to obtain homogeneous transformation matrix $T_h$ as $T_h = [T_{h11} \ T_{h12} \ T_{h13} \ x_1; T_{h21} \ T_{h22} \ T_{h23} \ x_3; T_{h31} \ T_{h32} \ T_{h33} \ x_5; 0 \ 0 \ 0 \ 1]$ where $T_{h11} = \cos(y_5)\cos(y_3)$, $T_{h12} = \cos(y_5)\sin(y_3)\sin(y_1) - \sin(y_5)\cos(y_3)$, $T_{h13} = \cos(y_3)\sin(y_3)\cos(y_1)$, $T_{h21} = \sin(y_5)\cos(y_3)$ $T_{h22} = \sin(y_5)\sin(y_3)\sin(y_1) + \cos(y_5)\cos(y_3)$, $T_{h23} = \sin(y_5)\sin(y_3)\cos(y_1) - \cos(y_5)\sin(y_1)$, $T_{h31} = -\sin(y_3)$, $T_{h32} = \cos(y_3)\sin(y_1)$, $T_{h33} = \cos(y_3)\cos(y_1)$.

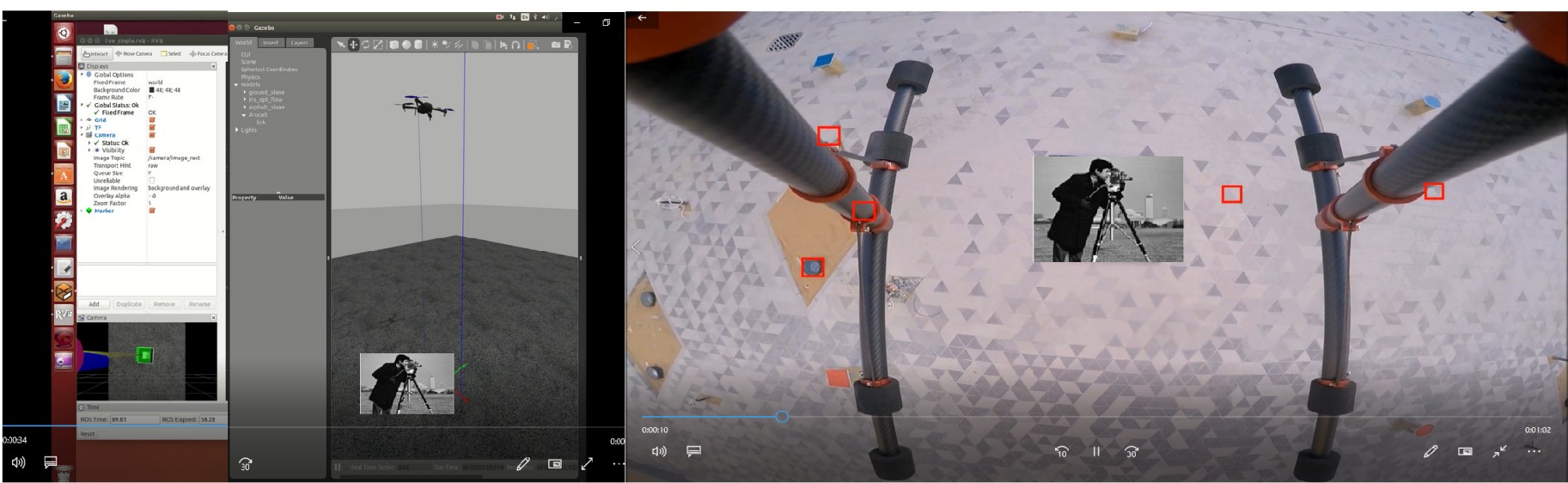

**Figure 14.** The image of the real scene of the flying vehicle interacting with target object.

Applying known constant transformation matrix $T_c^v$, the camera position to the ground $T_c^g$ can also be calculated. The desired camera velocity signals $v_T$ are obtained from camera velocities $v_c$ by using a transformation matrix linking the camera to vehicle $T_c^v$. Then, the visual tracking control laws $u_1$, $u_2$, $u_3$ and $u_4$ with the virtual control inputs $u_{tx}$ and $u_{ty}$ are implemented to force the quadrotor to the desired position and velocity signals of the attached camera. The tests are conducted for four cases. In our first case, it is assumed that the vehicle dynamics and inputs are free from uncertainty. In the second case, it is assumed that the vehicle inputs are subjected to unknown random noise uncertainty.

In the third case, the six acceleration measurements of the vehicle's dynamics are associated with unknown random noise uncertainty. The final case considers that the camera velocities are also subjected to unknown random noise uncertainty. Let us first test the proposed MWT based visual tracking system on the given quadrotor aerial vehicle with case 1. Figures 15–20 show the evaluation results of the proposed wavelet-based design for the vehicle with known uncertainty. Figures 15 and 16 present the generated inputs.

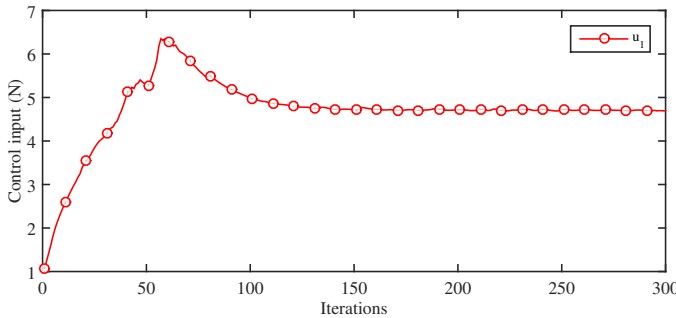

**Figure 15.** The input $u_1$ using wavelet based visual tracking system with Case 1.

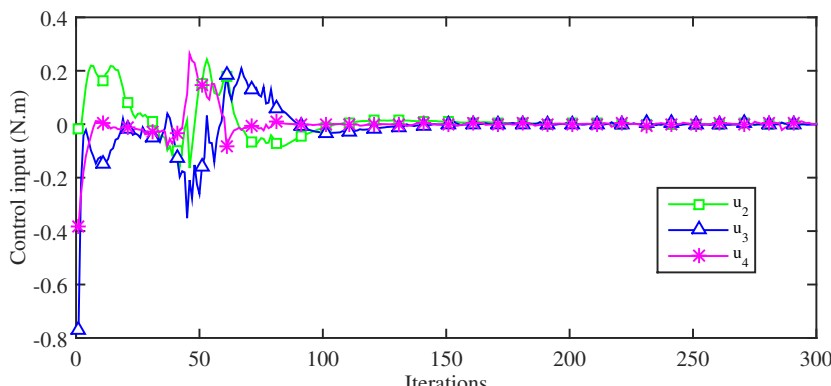

**Figure 16.** The control inputs $u_2$, $u_3$, $u_4$ with Case 1.

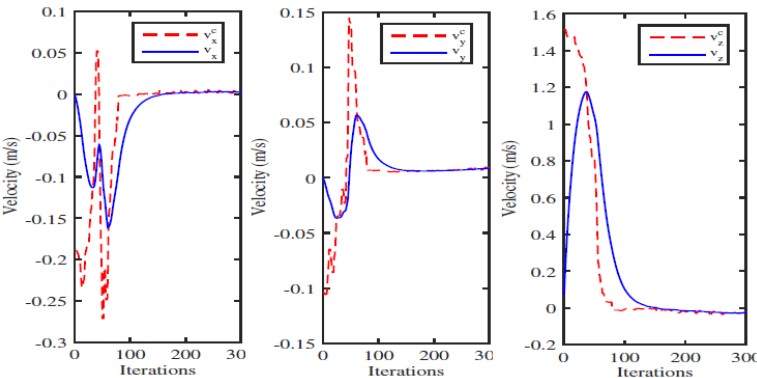

**Figure 17.** The desired and measured linear velocities in $x$, $y$, $z$ direction with Case 1.

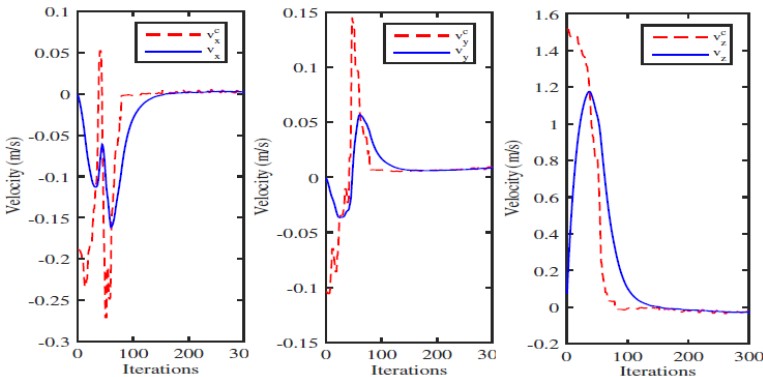

**Figure 18.** The desired and measured angular velocities in $x$, $y$, $z$ direction with Case 1.

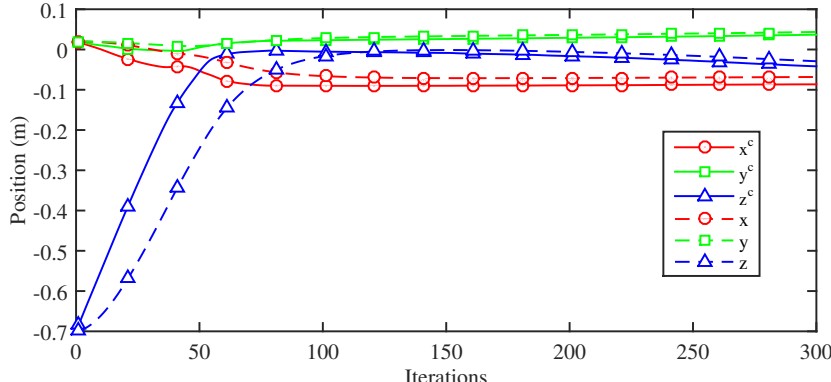

**Figure 19.** The desired and measured linear position with Case 1.

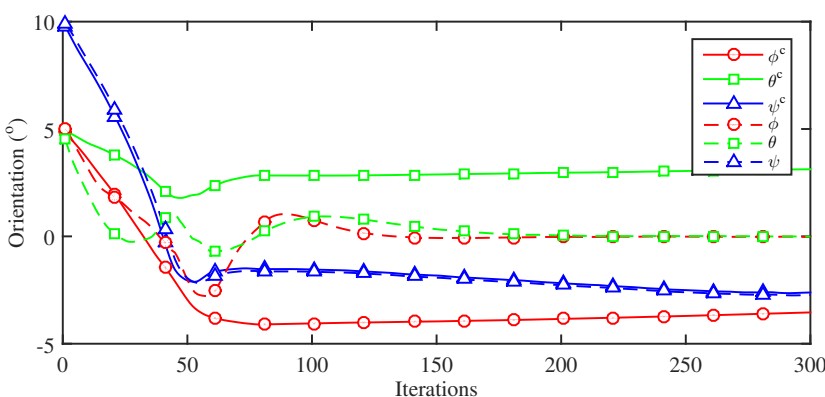

**Figure 20.** The desired and measured angular position with Case 1.

The evolution results of the desired and measured linear and angular velocities of the vehicle are given in Figures 17 and 18, respectively. The profiles of the desired and measured linear and angular positions of the vehicle are given in Figures 19 and 20, respectively. In view of these results, we can see that the measured position and velocity signals can track the desired position and velocity signals of the camera asymptotically. We now evaluate the performance of the wavelet-based tracking system in the presence of uncertainty. The design is tested with random Gaussian noise uncertainty associated with the thruster input and control inputs signals. Specifically, the vehicle input $u_1$ is contaminated by Gaussian noise uncertainty with mean $\mu_n = 0$ and variance $\sigma_n = 1$. The vehicle moments $u_2$, $u_3$ and $u_4$ are also contaminated by Gaussian noise uncertainty with $\mu_n = 0$ and $\sigma_n = 0.001$. For fair comparison, all design parameters are kept similar to our previous test. The evaluation results of the proposed design with the unknown noisy inputs present in Figures 21 and 22. Figures 22 and 23 show the profile of the thruster input and control inputs, respectively.

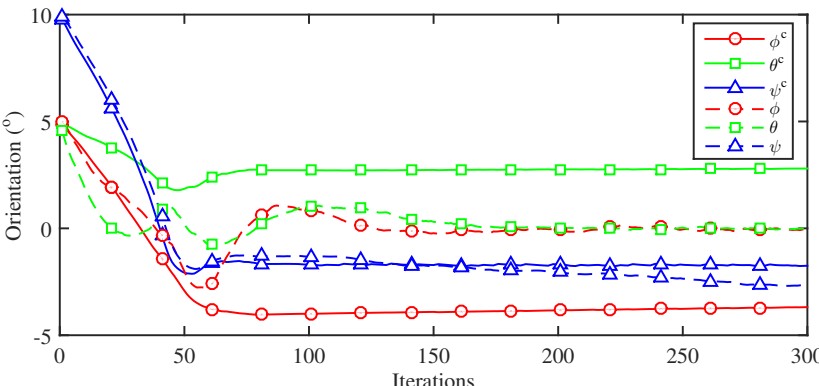

**Figure 21.** The desired and measured angular position in $x$, $y$, and $z$-direction with Case 2.

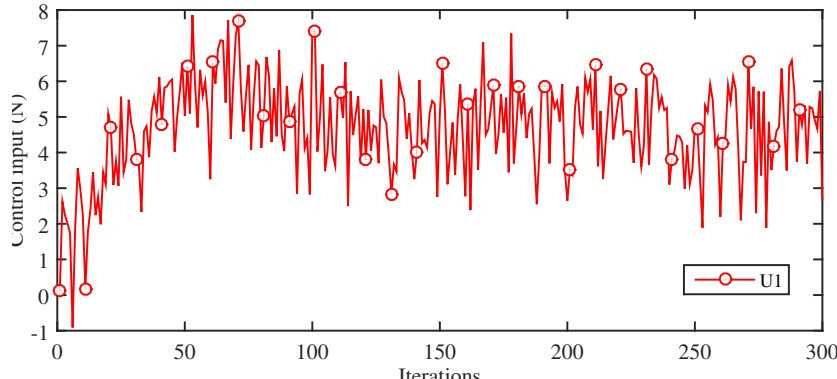

**Figure 22.** The thruster input contaminated with uncertainty with Case 2.

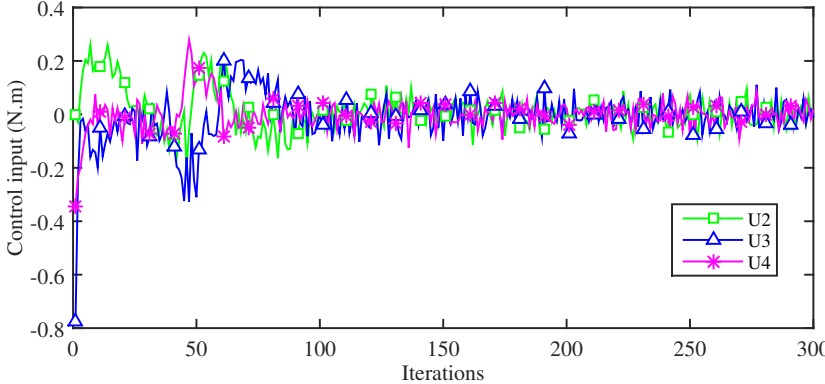

**Figure 23.** The control input when contaminated with uncertainty with Case 2.

The evolution results of the desired and measured linear velocities of the vehicle with noisy control inputs are shown in Figure 24. The desired and measured angular velocities of the vehicle are depicted in Figure 25. The desired and measured linear and angular position profiles are presented in Figures 21–26, respectively. Notice from these results that the system remains stable and ensures asymptotic stability with small oscillation with the linear and angular velocity states in the presence of uncertainty associated with the inputs. We now evaluate the proposed design with the presence of a random Gaussian noise uncertainty associated with the acceleration dynamics of the given quadrotor vehicle. The vehicles acceleration measurements are contaminated with Gaussian noise with parameters $\{\mu_n = 0, \sigma_n = 1\}$ for linear acceleration dynamics and $\{\mu_n = 0, \sigma_n = 0.001\}$ for angular acceleration dynamics.

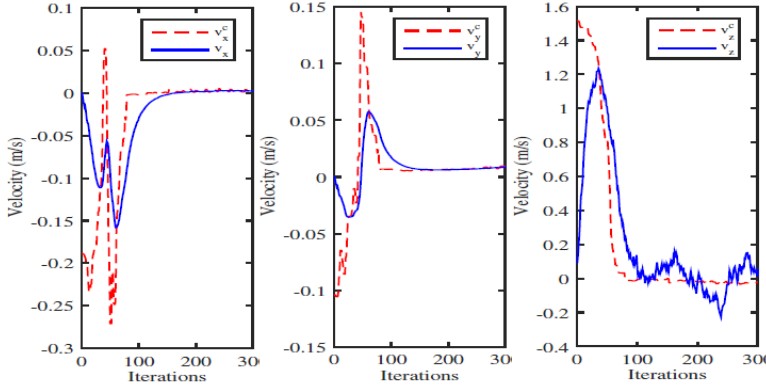

**Figure 24.** The desired and measured linear velocities in $x$, $y$, $z$-direction with Case 2.

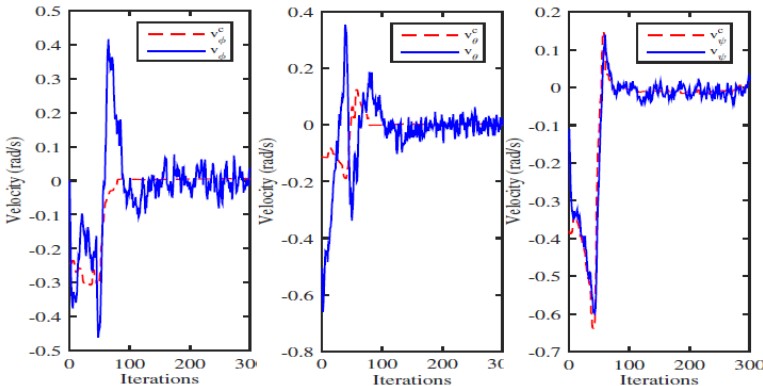

**Figure 25.** The desired and measured angular velocities in $x$, $y$, $z$-direction with Case 2.

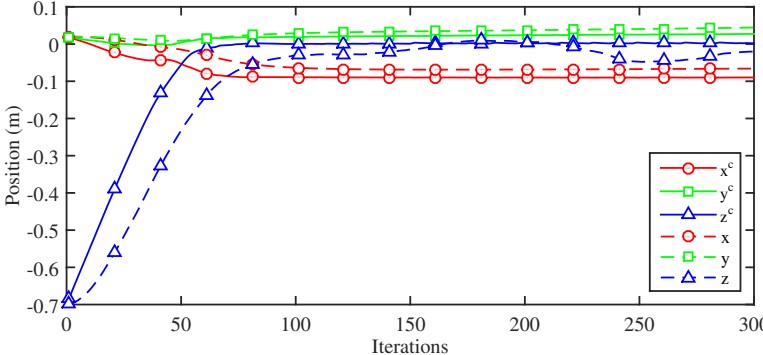

**Figure 26.** The desired and measured linear position in $x$, $y$ and $z$-direction with Case 2.

All other design parameters remain similar to our previous tests. Figures 27–32 depict the evaluation results with the presence of uncertain noisy acceleration measurement dynamics. Figures 27 and 28 show the input profiles of the vehicle with the contaminated linear and angular acceleration measurement uncertainty to iteration numbers. The desired and measured linear velocities of the vehicle are shown in Figure 29. Figure 30 depicts the profiles of the desired and measured angular velocities with respect to the iteration numbers. Figures 31 and 32 present the profiles of the desired and measured linear and angular positions of the vehicle with the noisy acceleration measurement uncertainty, respectively.

These results show that the wavelet based visual tracking system can ensure asymptotic tracking property of the error states of the vehicle even with the presence of very high noisy acceleration measurement dynamics. Finally, we examine the robustness of the proposed wavelet based visual tracking system on the given quadrotor aerial vehicle with the presence of uncertainty associated with the linear and angular velocities of the camera.

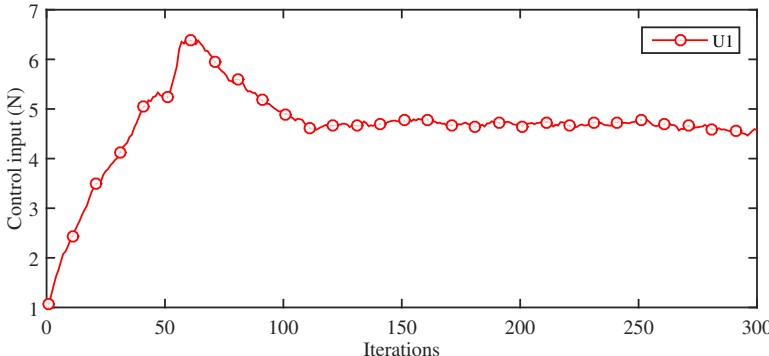

**Figure 27.** The thruster input with acceleration measurement uncertainty with Case 3.

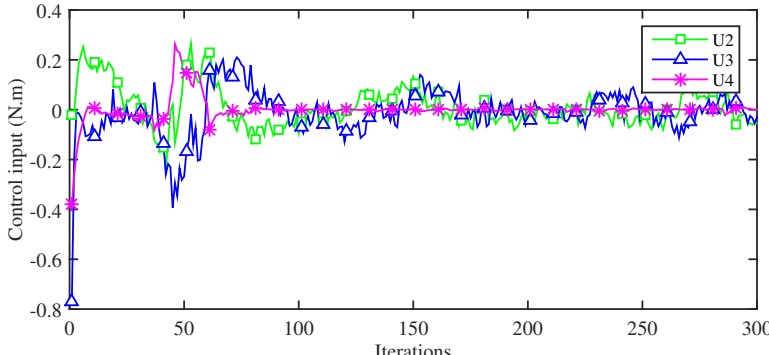

**Figure 28.** The control inputs with acceleration measurement uncertainty with Case 3.

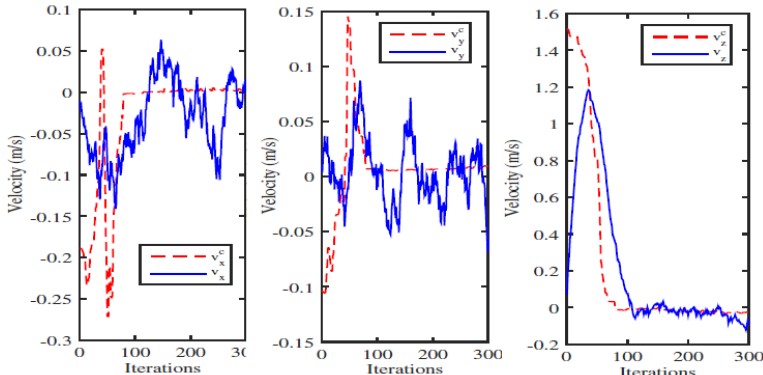

**Figure 29.** The desired and measured linear velocity in $x$, $y$, $z$-direction with noisy acceleration measurement uncertainty with Case 3.

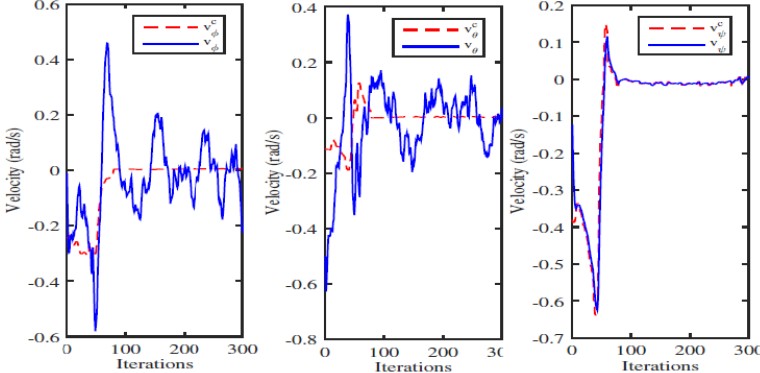

**Figure 30.** The desired and measured angular velocity in $x$, $y$, $z$-direction with Case 3.

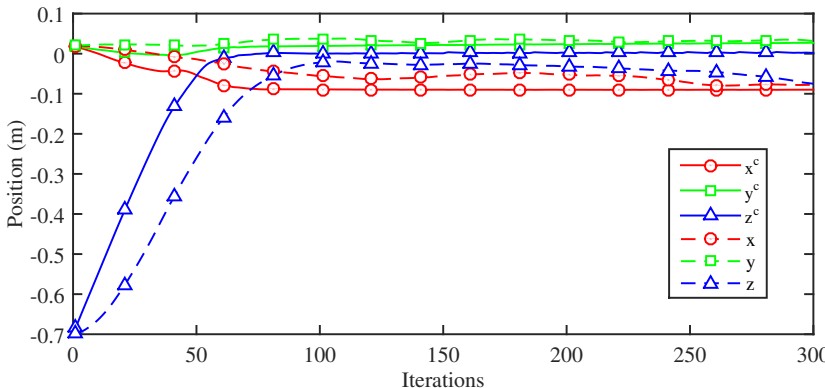

**Figure 31.** The desired and measured linear position in $x$, $y$, and $z$-direction with Case 3.

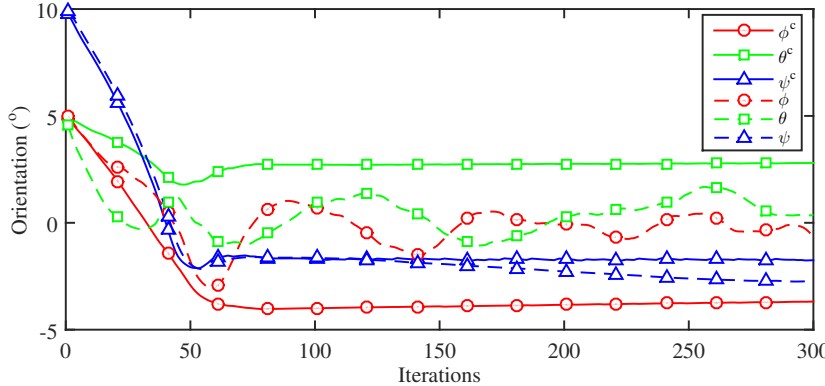

**Figure 32.** The desired and measured angular position in $x$, $y$, and $z$-direction with Case 3.

Unlike the previous cases, this test affects both camera view and visual tracking process to affect the control system of the vehicle significantly. The goal is to examine the robustness of the design with the presence of uncertainty associated with the linear and angular camera velocities. To do that, a Gaussian noise uncertainty is added with the computed camera velocities. The linear velocities of the camera is contaminated with Gaussian noise with parameters $\{\mu_n = 0, \sigma_n = 0.04\}$. The angular velocities of the camera are contaminated with Gaussian noise with parameters $\{\mu_n = 0, \sigma_n = 0.01\}$. For fair comparison, all other design parameters and test scenarios remain the same as our previous cases. The tested results with the presence of uncertainty associated with the linear and angular camera measurement velocities depict in Figures 33 and 34 show. Despite the uncertainty associated with camera velocities, the MWT based visual tracking process managed to bring the camera view closed to the desired view as depicted in Figure 34. Figures 35 and 36 show the control inputs of the vehicle when the linear and angular camera velocities are contaminated with uncertainty. Figure 37 shows the desired noisy linear camera velocities and measured linear velocities of the vehicle with Case 4. Figure 38 depicts the desired noisy angular camera velocities and measured angular velocities of the vehicle with Case 4. The linear position states of the vehicle and camera are given in Figure 39 with Case 4. Figure 33 presents angular position states of the vehicle and camera with Case 4. Notice from these results that the vehicle remains stable and maintains good convergence accuracy even with the presence of large uncertainty associated with the desired linear and angular camera velocity signals.

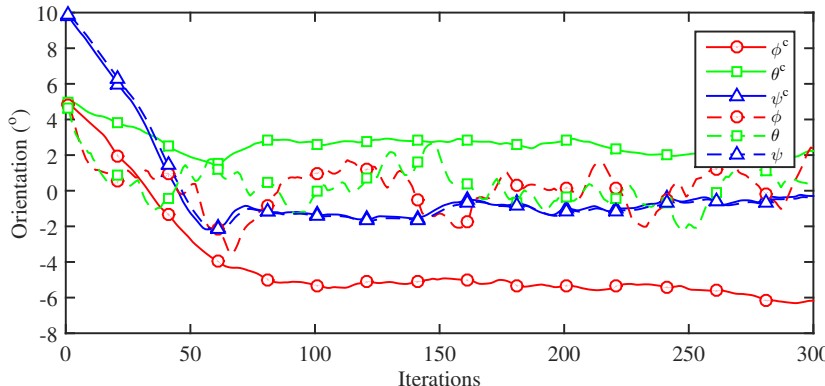

**Figure 33.** The desired camera and measured angular position in $x$, $y$, and $z$-direction with the presence of uncertainty in camera velocities.

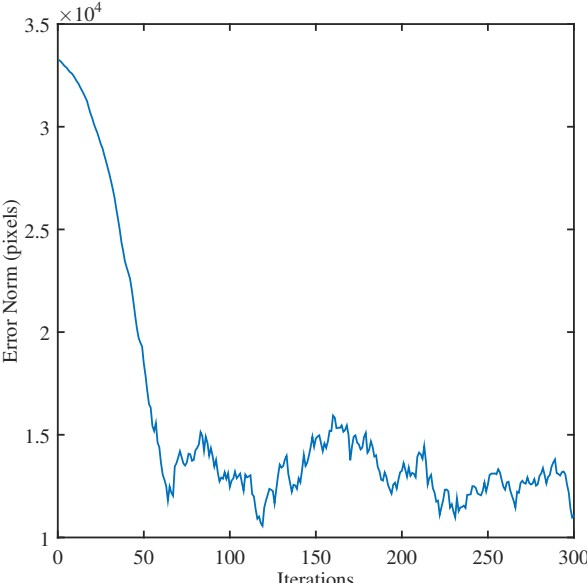

**Figure 34.** The profile of the error norm with the presence of the uncertainty associated with the linear and angular camera velocities with Case 4.

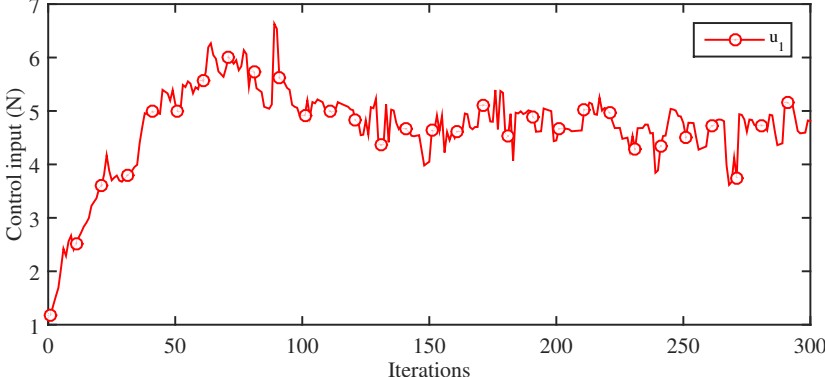

**Figure 35.** The thruster input profile with the presence of uncertainty with the linear and angular camera velocities with Case 4.

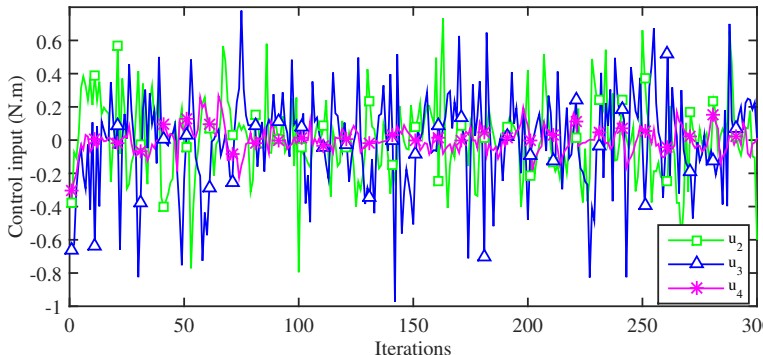

**Figure 36.** The control inputs profile with the presence of uncertainty with the linear and angular camera velocities with Case 4.

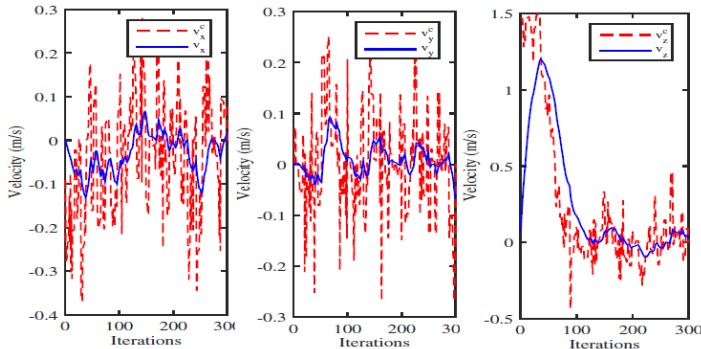

**Figure 37.** The desired noisy camera and measured linear velocity of the vehicle in *x*-direction with Case 4.

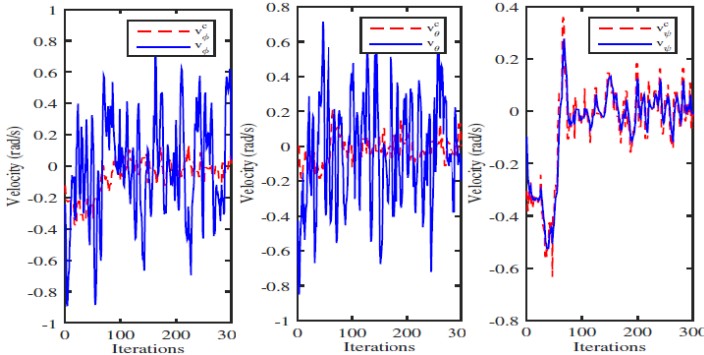

**Figure 38.** The desired noisy camera and measured angular velocity in *x*, *y*, *z*-direction with Case 4.

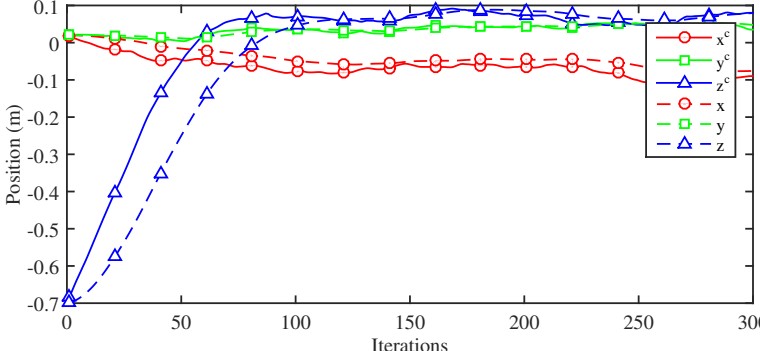

**Figure 39.** The desired camera and measured linear position in *x*, *y* and *z*-direction with Case 4 with uncertainty associated with the camera velocities.



Remarks 1: The classical visual tracking process aims to detect and match the geometrical features in the image such as points, lines, and circles. However, this segment slows down the tracking process as it is computationally expensive. Moreover, adding an extra algorithm for 3D perception will result in a serious drawback in terms of convergence speed. This is seen as the main obstacle of the tracking task in many cases. To overcome these problems, MWT based techniques are developed to eliminate the need for a visual tracking process. The main idea is to use features that can be directly derived from the images without processing, if not using the whole image as the visual signal.

Remark 2: In our future work, adaptive learning based visual tracking system for aerial vehicle will be designed to deal with uncertainty associated with the unceratinty along the line of the method proposed in [46–48]. The future works will also be involved in testing the design on a quadrotor aerial vehicle.

## 4. Conclusions

In this paper, we have developed a wavelet-based image-guided visual tracking system for an unmanned aerial vehicle. The visual tracking system has been designed by using wavelet coefficients of the half and full image. The design developed a multiresolution interaction matrix to relate the time-variation of the wavelet coefficients with the velocity of the vehicle and controller. The proposed wavelet-based MWT design was implemented and evaluated on a virtual quadrotor aerial vehicle simulator. The evaluation results showed that the MWT based visual tracking system can achieve accuracy and efficiency even without using an image processing unit as opposed to a classical visual tracking mechanism. The tests show that the proposed MWT based visual tracking system can ensure the stability of the vehicle and guarantee good convergence accuracy even with the presence of uncertainty associated with the camera velocities, vehicle dynamics, and vehicle inputs.

**Author Contributions:** Conceptualization, S.I. and J.D.; methodology, S.I.; software, S.I. and H.M.; validation, S.I. and H.M.; formal analysis, S.I.; investigation, S.I. and J.D.; resources, S.I. and J.D.; data curation, S.I. and H.M.; writing—original draft preparation, S.I.; writing—review and editing, S.I.; visualization, S.I. and J.D.; supervision, S.I. and J.D.; project administration, S.I. and J.D.; funding acquisition, S.I. and J.D. All authors have read and agreed to the published version of the manuscript..

**Funding:** This research received no external funding.

**Conflicts of Interest:** The authors declare no conflict of interest.

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
