# Peer review of "Image Guided Visual Tracking Control System for Unmanned Multirotor Aerial Vehicle with Uncertainty†"

_robotics, doi:10.3390/robotics9040103_

Round 1

Reviewer 1 Report

The authors introduced wavelet-based image guided tracking control system for UAV with the presence of uncertainty. The proposed wavelet-based design is evaluated on a virtual quadrotor aerial vehicle system to illustrate its effectiveness on real system without using image processing unit.

    The idea is new and interesting in vision based autonomous tracking of aerial vehicle system. So, I recommend for accepting the paper in current form. Authors may include the following recommendation before final submission

Comments

-This reviewer suggests reducing the page of the manuscript.

-Author may clarify how the design parameters are chosen in page 8 for experimental evaluation.

- It is important that authors must include a separate remark on classical visual tracking system as opposed to wavelet-based design.

-Why did author only use single camera?

-Authors can write a separate remark on wavelet based visual tracking system.

- How did author select Lyapunov function in equation (37)

-Author should include the future works in separate remark.

Author Response

Reviewer 1

The authors introduced wavelet-based image guided tracking control system for UAV with the presence of uncertainty. The proposed wavelet-based design is evaluated on a virtual quadrotor aerial vehicle system to illustrate its effectiveness on real system without using image processing unit.

    The idea is new and interesting in vision based autonomous tracking of aerial vehicle system. So, I recommend for accepting the paper in current form. Authors may include the following recommendation before final submission

Thank you very much for reviewing our paper. Thank you very much for your comments.

 We have revised the paper according to your suggestions.

Comments

-This reviewer suggests reducing the page of the manuscript.

As you suggested, we reduce the page in our revised submission.

-Author may clarify how the design parameters are chosen in page 8 for experimental evaluation.

The design parameters are chosen by trial error and search method.

- It is important that authors must include a separate remark on classical visual tracking system as opposed to wavelet-based design.

As you suggested, we provide a separate remark in our revised submission

-Why did author only use single camera?

Then single camera is chosen as its field of view can cover the target image  

-Authors can write a separate remark on wavelet based visual tracking system.

As you suggested, we provide a separate remark in our revised submission.  

- How did author select Lyapunov function in equation (37)

The experience plays a significant role in choosing Lyapunov function candidate. The trial and error technique is required to select an appropriate function candidate.

-Author should include the future works in separate remark.

As you suggested, we provide a separate remark in our revised submission.  

Reviewer 2 Report

The english needs to be improved. In general too many figures; should instead be handled by plotting all velocity components together, and using subplots.Some of the pictures have y-axis text chopped, and other figures are poorly scaled. In general an interesting paper that needs improvements.

Author Response

Reviewer 2

Comments and Suggestions for Authors

The english needs to be improved. In general too many figures; should instead be handled by plotting all velocity components together, and using subplots. Some of the pictures have y-axis text chopped, and other figures are poorly scaled. In general an interesting paper that needs improvements.

---------Thank you very much for reviewing our paper. Thank you very much for your thoughts and suggestion on our earlier submission. We have revised the paper according to your suggestions. Specifically, we plot velocity figures together in revised version of the manuscript. As you suggested, we have checked the manuscript to make sure axis are readable. We also went through whole manuscript couple of times to make sure English’s are improved. 

Reviewer 3 Report

This paper presents a tracking control system for unmanned multirotor aerial vehicle based on wavelet image. The authors claim that the proposed visual tracking system is effective in the presence of uncertainty and does not require image processing task. Generally speaking, the paper is not written well and more work have to be conducted to be considered acceptable.  Main issues are listed as follows.

  1. The author did not show the configuration of the camera of the UAV, whether it is forward looking or downward looking?
  2. The experiments are conducted in a simulator, and it is easy to designate the desired position and velocities of the UAV, but how to designate the desired positions and velocities on a real UAV?
  3. The result is not convincing as the authors only conducted the experiments in simulation, the authors are encouraged to conduct experiments on real UAVs so that the proposed algorithm and result is more persuasive.
  4. The scene in Figure 9-11 is not the real scene of a UAV flying, and images of real scene of UAV flying are better to illustrate principle of the MWT based visual tracking system.
  5. The figures in the paper can be placed in a more compact fashion as current layout of figures influences readability of the paper.

Author Response

Reviewer 3

Comments and Suggestions for Authors

This paper presents a tracking control system for unmanned multirotor aerial vehicle based on wavelet image. The authors claim that the proposed visual tracking system is effective in the presence of uncertainty and does not require image processing task. Generally speaking, the paper is not written well and more work have to be conducted to be considered acceptable.  

-------------Thank you very much for reviewing our paper. Thank you very much for your thoughts and suggestion on our earlier submission. We have revised the paper according to your suggestions.

Main issues are listed as follows.

  1. The author did not show the configuration of the camera of the UAV, whether it is forward looking or downward looking?

-----As you suggested, we show configuration of the camera of the vehicle. The vehicle is equipped with a downward camera attached with the center of the vehicle. The camera is equipped with the vehicle such that it can look downward with relation to the vehicle. The target is synchronized in such a way that its axes are parallel to the local plane North East axes. 

  1. The experiments are conducted in a simulator, and it is easy to designate the desired position and velocities of the UAV, but how to designate the desired positions and velocities on a real UAV?

----Thank you very much for your comments and your thoughts. The proposed MWT design uses the image of the known target. Therefore, the desired position and velocity signals for the vehicle is assigned based on the position and velocity of the target image.  

  1. The result is not convincing as the authors only conducted the experiments in simulation, the authors are encouraged to conduct experiments on real UAVs so that the proposed algorithm and result is more persuasive.

---Thank you very much for your comments. We will test the wavelet based visual tracking system on quadrotor aerial vehicle in our future works accordingly.

  1. The scene in Figure 9-11 is not the real scene of a UAV flying, and images of real scene of UAV flying are better to illustrate principle of the MWT based visual tracking system.

---As you suggested we include the image of the real scene of vehicle flying interacting with target object.

  1. The figures in the paper can be placed in a more compact fashion as current layout of figures influences readability of the paper.

---Thank you very much for your suggestion. As you recommended, some of the figures are placed in a compact form in our revised submission.

Round 2

Reviewer 3 Report

1) The authors still did not conduct the experiments in real world;
2) The paper layout is not revised very well, as some figures are placed in reference parts.
3) The target image in Figure 14 seems to be pasted into the real scene by a photo-editing software, rather than embedded in the simulator Gazebo directly. This allows me to doubt the result of the simulation.

Author Response

Comments and Suggestions for Authors

1) The authors still did not conduct the experiments in real world;

-Thank you very much for your comments. As highlighted by remarks, the future work will be involved in testing the design on a quadrotor aerial vehicle. We are testing the proposed design on virtual and industrial quadrotor aerial vehicle where the vehicle is landing on fixed and moving target application. 

2) The paper layout is not revised very well, as some figures are placed in reference parts.

-As you suggested, we remove the figures from the reference parts
3) The target image in Figure 14 seems to be pasted into the real scene by a photo-editing software, rather than embedded in the simulator Gazebo directly. This allows me to doubt the result of the simulation.

--As you recommended, we had included the screen shot of the image from the evaluation. We are currently testing the design on  virtual simulator and industrial quadrotor vehicle for landing on fixed and moving target. Some tests on fixed target can be seen from the following link 

https://youtu.be/FkPCQbl0pjs https://youtu.be/IYrAm7kVWsQ  https://youtu.be/pipCMIXqXK0  https://youtu.be/bQM4U07YfiQ